# PURE: An Uncertainty-aware Recommendation Framework for Maximizing Expected Posterior Utility of Platform

## ABSTRACT

Commercial recommendation can be regarded as an interactive process between the recommendation platform and its target users. One crucial problem for the platform is how to make full use of its advantages so as to maximize its utility, i.e., the commercial benefits from recommendation. In this paper, we propose a novel recommendation framework which effectively utilizes the information of user uncertainty over different item dimensions[1] and explicitly takes into consideration the impact of display policy on user in order to achieve maximal expected posterior utility for the platform. We formulate the problem of deriving optimal policy to achieve maximal expected posterior utility as a constrained non-convex optimization problem and further propose an ADMM-based solution to derive an approximately optimal policy. Extensive experiments are conducted over data collected from a real-world recommendation platform and demonstrate the effectiveness of the proposed framework. Besides, we also adopt the proposed framework to conduct experiments with an intent to reveal how the platform achieves its commercial benefits. The results suggest that the platform should cater to the user's preference for item dimensions that the user prefers, while for item dimensions where the user is with high uncertainty, the platform can achieve more commercial benefits by recommending items with high utilities.

## 1 INTRODUCTION

Commercial recommendation systems have been widely applied among prevalent content distribution platforms such as YouTube, TikTok, Amazon and Taobao. During the interactive process on the recommendation platform, the users may find contents of their interests and avoid the information overload problem with the help of recommendation services. Meanwhile, the platform may gain commercial benefits from user behaviors on the platform such as clicks and purchases. As the platform may serve millions of users and can determine which contents to be recommended, it naturally has some advantages over individual user. Therefore, it would be crucial for the platform to make full use of its advantages in order to maximize the commercial benefits.

One typical advantage of the platform is its information advantage, i.e., they may collect plenty of information over users and items for conducting better recommendation. Typical state-of-the-art recommendation systems (Covington et al., 2016; Guo et al., 2017; Ren et al., 2019; Zhou et al., 2019) always take these information into consideration including user profiles, item features and historical interactions between users and recommended items. It is worth noting that information over item features is always directly incorporated into the recommendation models without considering that the user may be with different levels of uncertainty over different item dimensions (which can be regarded as different hidden attributes describing different high-order features of the item). For instance, when buying a new coat on the platform, a user may be sure that the logistics is very fast as she (he) has bought clothes from the same online store before (i.e., the user is with low uncertainty over the logistics). But she (he) may be uncertain about the quality of the coat since it is of the brand that she (he) does not know much about (i.e., the user is with high uncertainty over the quality). Thus, it would be crucial for the platform to figure out whether it is possible to leverage the user uncertainty over different item dimensions to maximize the platform utility, and if yes, how?

---

[1]Item dimensions: Typical state-of-the-art solutions for recommendation systems always encode each item as an embedding. The item dimensions refer to different dimensions of the item embedding, which can be explained as different high-order features.

Actually, with consideration of the user uncertainty over different item dimensions, we would show that more commercial benefits can be gained from the item dimensions with higher uncertainty.

Another advantage of the platform is that it owns the capacity of determining which items to display for the users and thus may affect the users' behaviors. It has been proved by lots of works (Kamenica & Gentzkow, 2011; Immorlica et al., 2019; Abdollahpouri & Mansoury, 2020) that the display signal itself would highly affect users' behaviors, and affected behaviors would apparently result in different benefits for the platform. Regarding the recommendation as a game between the platform and the users, it is possible for the platform to achieve more commercial benefits from the game by taking a proper display (recommendation) policy. However, though there are works to explore the impact of recommendation policies, it is still not well-studied in recommendation area how to explicitly model and exploit the impact of the display policy over users.

In this paper, we propose an uncertainty-aware expected **P**osterior **U**tility maximization framework for **RE**commendation platforms (denoted as **PURE** in short). We take both the two previously mentioned factors, i.e., user uncertainty over different item dimensions and influence of display policy over the user, into account and introduce a generic utility function which can be flexibly adjusted for different real-world scenarios. Then, we formulate the problem of maximizing expected posterior utility for the platform as a constrained non-convex optimization problem, and correspondingly propose a solution based on Alternating Direction Method of Multipliers (ADMM, Boyd et al. (2011)) to derive the approximately optimal policy. To verify the effectiveness of the proposed framework, extensive experiments are conducted over data collected from a real-world recommendation platform. Furthermore, we also provide practical insights derived from carefully designed experiments and empirically reveal how the platform utilizes its information advantage to achieve more commercial benefits, which may help to better understand and conduct commercial recommendation.

## 2 RELATED WORK

Existing state-of-the-art recommendation systems (Zhou et al., 2018; Pi et al., 2019; Qu et al., 2016) mainly try to make full use of the information advantage of the platform. These works take these information into consideration including user profiles, item features, contextual information and historical interactions between users and recommended items. Typically, some works (Qu et al., 2016; Zhou et al., 2018; Li et al., 2019) focus on how to achieve better feature interactions or conduct better user interest modeling, while some works (Ren et al., 2019; Pi et al., 2019) may pay more attention to utilizing extremely long sequential interactive information. However, most of them ignore the existence of user uncertainty over different item dimensions, which might be crucial to conduct better commercial recommendation.

In the research area to explore the display influence to the information receiver, Bayesian Persuasion (Kamenica & Gentzkow, 2011) is one of the most crucial works, which theoretically proves that the information sender may benefit from displaying proper information to the receiver. Some works (Immorlica et al., 2019; Mansour et al., 2016) follow this idea and strive to incentivize exploration via information asymmetry in scenarios such as recommendation. In another research direction that try to develop Reinforcement Learning (RL) based solutions for recommendation scenarios, a series of works (Dulac-Arnold et al., 2015; Zhao et al., 2018; Chen et al., 2019) model the recommendation process as a Markov Decision Process (MDP) and maximize the long-term reward via utilizing learned sequential patterns, which can also be regarded as taking the display (recommendation) influence into consideration to some extent.

## 3 METHODOLOGY

### 3.1 OPTIMAL POLICY FOR MAXIMIZING PLATFORM'S EXPECTED POSTERIOR UTILITY

From the perspective of the platform, the optimal recommendation policy is the one with maximal expected utility (i.e., maximal expected commercial benefits). As mentioned before, the influence of display policy over users can not be ignored as it would highly affect the commercial benefits of the platform. In this paper, taking the impact of display policy on users into consideration, we formulate the platform's optimal policy $\boldsymbol{\pi}^u$ for user $u$ over a given item set $\mathbb{I}$ as follows.

$$\boldsymbol{\pi}^u = \underset{\boldsymbol{\pi}}{argmax} \sum_{i \in \mathbb{I}} \boldsymbol{\pi}_i U_u(i|display; \boldsymbol{\pi}), s.t., \forall i \in \mathbb{I}, \boldsymbol{\pi}_i \geq 0 \ \ and \ \sum_{i \in \mathbb{I}} \boldsymbol{\pi}_i = 1, \quad (1)$$

where $U_u(i|display; \boldsymbol{\pi})$ is the posterior utility of recommending item $i$ to user $u$ with consideration of the influence of display policy $\boldsymbol{\pi}$. With this formulation, the remaining problem is how to model

the posterior utility properly. In the following, we illustrate two reasonable assumptions in detail, which make it possible to model the posterior utility with consideration of the user uncertainty over different item dimensions as well as the influence of display policy over user.

As discussed before, it would be crucial to explicitly consider the user uncertainty over different item dimensions to conduct recommendation. For a given user, we assume that the representation for an item is sampled from a multivariate Gaussian distribution and adopt the variances to describe the user uncertainty over different item dimensions, which is formulated as the following assumption.

**Assumption 1** (Assumption of uncertainty (correlation) over different item dimensions). *For a user $u$, the representation of item $i$ is sampled from a $n$-dimension multivariate Gaussian distribution $\mathcal{N}(\boldsymbol{\mu}_{u,i}, \boldsymbol{\Sigma}_{u,i})$, i.e., the probability density function of the representation is:*

$$p_{u,i}(\boldsymbol{x}) = \frac{1}{\sqrt{(2\pi)^n |\boldsymbol{\Sigma}_{u,i}|}} e^{-\frac{1}{2}(\boldsymbol{x}-\boldsymbol{\mu}_{u,i})^T \boldsymbol{\Sigma}_{u,i}^{-1}(\boldsymbol{x}-\boldsymbol{\mu}_{u,i})}, \tag{2}$$

*where $\boldsymbol{x} \in \mathbb{R}^n$, $\boldsymbol{\mu}_{u,i}$ and $\boldsymbol{\Sigma}_{u,i}$ denote the mean vector and the covariance matrix respectively.*

The covariance matrix can be decomposed as $\boldsymbol{\Sigma}_{u,i} = \boldsymbol{D}_{u,i} \boldsymbol{C}_{u,i} \boldsymbol{D}_{u,i}$, where $\boldsymbol{D}_{u,i}$ is the diagonal standard deviation matrix and $\boldsymbol{C}_{u,i}$ is the correlation matrix (see Barnard et al. (2000) for more information). Thus, the covariance matrix can depict the user uncertainty over different item dimensions (with diagonal standard deviation matrix $\boldsymbol{D}_{u,i}$) as well as the correlation between different item dimensions (with correlation matrix $\boldsymbol{C}_{u,i}$). Note that we provide a practical method to gain $\boldsymbol{\mu}_{u,i}$ and $\boldsymbol{\Sigma}_{u,i}$ in Section 4.1 while any other reasonable approach to get $\boldsymbol{\mu}_{u,i}$ and $\boldsymbol{\Sigma}_{u,i}$ can be applied.

From the perspective of users, they may try to understand the display policy of the platform from the interactive process. When an item $i$ is recommended to a user $u$, the user may consider the corresponding display probability, which could influence his behavior. One reasonable assumption is that the probability of displaying item $i$ from the perspective of user $u$ is linear to the similarity between the item representation $\boldsymbol{x}$ and the representations of historical recommended items. Without loss of generality, we formulate this assumption as follows.

**Assumption 2** (Assumption of the influence of the display policy over user). *Given display policy $\boldsymbol{\pi}$ and item representation $\boldsymbol{x}$, the probability of recommending item $i$ to user $u$ from the user's perspective is:*

$$p_{u,i}(display|\boldsymbol{x};\boldsymbol{\pi}) = \Phi(a\boldsymbol{v}_u^T\boldsymbol{x} + b), \tag{3}$$

*where $display$ denotes the event of displaying (recommending) the corresponding item to the target user, $a$ and $b$ are scale hyper-parameters, $a > 0$, $\Phi(z) = \int_{-\infty}^{z} \frac{1}{\sqrt{2\pi}} e^{-\frac{x^2}{2}} dx$ and $\boldsymbol{v}_u = \mathbb{E}_{i\sim\boldsymbol{\pi}}(\boldsymbol{\mu}_{u,i}) = \sum_{i\in\mathbb{I}} \boldsymbol{\pi}_i \boldsymbol{\mu}_{u,i}$.*

Note that $\Phi$ is the cumulative distribution function of standard normal distribution, which is widely adopted to map the input into range $[0, 1]$ in many models such as probit model, and $\boldsymbol{v}_u$ takes the expected value of item representations w.r.t. the display policy $\boldsymbol{\pi}$ over item set $\mathbb{I}$. Thus, with $a > 0$, higher similarity (calculated by inner product) between the current item representation $\boldsymbol{x}$ and the expected representation of display items would lead to higher likelihood, which is reasonable as discussed before.

So far, we have presented two crucial assumptions adopted in our framework. In the following, we introduce the utility function and derive the formula of posterior utility. To ease the illustration, we denote the utility function of recommending item $i$ to user $u$ given sampled item representation $\boldsymbol{x}$ as $f(\boldsymbol{w}_u, \boldsymbol{x})$, where $\boldsymbol{w}_u$ is the embedding of user side typically and $f(\boldsymbol{w}_u, \boldsymbol{x})$ depicts the utility (benefits) that the platform can gain from recommending item $i$ to user $u$ with item representation $\boldsymbol{x}$. For instance, when we regard the click trough rate (CTR) as the platform's utility, one simple way is to model $f$ as the inner product of $\boldsymbol{w}_u$ and $\boldsymbol{x}$ where $\boldsymbol{w}_u$ can be regarded as the user preference vector of user $u$. Note that the function $f$ can be flexibly adjusted to fit different requirements of different scenarios. For example, when we try to maximize the gross merchandise volume (GMV) in a recommendation scenario, $f$ can be defined as the product of the corresponding CTR, conversion rate (CVR) and the item price.

Now we can combine the two assumptions and the utility function $f$ to derive the formula of posterior utility. By adopting the Bayes' theorem and the law of total probability, we present the formula of posterior utility of recommending item $i$ to user $u$ as follows.

$$U_u(i|display;\boldsymbol{\pi}) = \int_{\mathbb{R}^n} f(\boldsymbol{w}_u, \boldsymbol{x}) p_{u,i}(\boldsymbol{x}|display;\boldsymbol{\pi}) \, d\boldsymbol{x} \tag{4}$$

$$= \int_{\mathbb{R}^n} f(\boldsymbol{w}_u, \boldsymbol{x}) p_{u,i}(\boldsymbol{x}) \frac{p_{u,i}(display|\boldsymbol{x};\boldsymbol{\pi})}{p_{u,i}(display;\boldsymbol{\pi})} \, d\boldsymbol{x} \tag{5}$$

Equation 5 provides a way to model the posterior utility of the platform taking into account both the user uncertainty over different item dimensions and the influence of display policy over user. However, it is still challenging to derive the optimal policy $\boldsymbol{\pi}^u$ for given user $u$ as the right part of Equation 5 is not a close-form expression (which makes it intractable to calculate the exact value of the posterior utility).

## 3.2 POSTERIOR UTILITY DERIVATION FOR LINEAR AND NON-LINEAR UTILITY FUNCTIONS

In this section, we present how to compute (estimate) the value of posterior utility for both linear and non-linear utility functions. To derive an effective calculation formula of $U_u(i|display;\boldsymbol{\pi})$ for the case with linear utility function, we first introduce a lemma in the following.

**Lemma 1.** $\int_{-\infty}^{+\infty}(c'x+d')\Phi(a'x+b')\mathcal{N}(x|0,1)\,dx = \frac{a'c'}{\sqrt{2\pi(a'^2+1)}}e^{-\frac{b'^2}{2a'^2+2}} + d'\Phi(\frac{b'}{\sqrt{1+a'^2}}), \forall a', b',$
$c', d' \in \mathbb{R}$.

The detailed proof of Lemma 1 is provided in Appendix A.1 due to page limits. Lemma 1 reveals the fact that the definite integral $\int_{-\infty}^{+\infty}(c'x+d')\Phi(a'x+b')\mathcal{N}(x|0,1)\,dx$ can be computed directly, which is a crucial property to derive an effective calculation formula of $U_u(i|display;\boldsymbol{\pi})$ for the case with linear utility function.

By utilizing Lemma 1, the following corollary shows that $p_{u,i}(display;\boldsymbol{\pi})$ can be straightforwardly calculated if the covariance matrix is positive definite.

**Corollary 1.** $p_{u,i}(display;\boldsymbol{\pi}) = \Phi(\frac{a\boldsymbol{v}_u^T\boldsymbol{\mu}_{u,i}+b}{\sqrt{1+a^2\boldsymbol{v}_u^T\boldsymbol{\Sigma}_{u,i}\boldsymbol{v}_u}})$, if $\boldsymbol{\Sigma}_{u,i}$ is positive definite.

Due to page limits, the detailed proof of Corollary 1 is also presented in Appendix A.2. With Lemma 1 and Corollary 1, we can derive the calculation formula of $U_u(i|display;\boldsymbol{\pi})$ if the utility function is linear, which is illustrated in detail as follows.

**Corollary 2.** $U_u(i|display;\boldsymbol{\pi}) = \boldsymbol{w}_u^T\boldsymbol{\mu}_{u,i} + \frac{a\boldsymbol{v}_u^T\boldsymbol{\Sigma}_{u,i}\boldsymbol{w}_u}{\sqrt{2\pi(a^2\boldsymbol{v}_u^T\boldsymbol{\Sigma}_{u,i}\boldsymbol{v}_u+1)}\Phi(\frac{a\boldsymbol{v}_u^T\boldsymbol{\mu}_{u,i}+b}{\sqrt{a^2\boldsymbol{v}_u^T\boldsymbol{\Sigma}_{u,i}\boldsymbol{v}_u+1}})}e^{-\frac{(a\boldsymbol{v}_u^T\boldsymbol{\mu}_{u,i}+b)^2}{2(a^2\boldsymbol{v}_u^T\boldsymbol{\Sigma}_{u,i}\boldsymbol{v}_u+1)}}$,

if $f(\boldsymbol{w}_u, \boldsymbol{x}) = \boldsymbol{w}_u^T\boldsymbol{x}$ and $\boldsymbol{\Sigma}_{u,i}$ is positive definite.

*Proof.* If $\boldsymbol{\Sigma}_{u,i}$ is positive definite, $\boldsymbol{\Sigma}_{u,i}^{-1}$ is also positive definite. Then we can conduct Cholesky decomposition: $\boldsymbol{\Sigma}_{u,i}^{-1} = \boldsymbol{A}_{u,i}^T\boldsymbol{A}_{u,i}$, where $\boldsymbol{A}_{u,i}$ is an upper triangular matrix with real and positive diagonal entries. Thus, we have:

$$\int_{\mathbb{R}^n} f(\boldsymbol{w}_u, \boldsymbol{x}) p_{u,i}(\boldsymbol{x}) p_{u,i}(display|\boldsymbol{x};\boldsymbol{\pi}) \, d\boldsymbol{x} \tag{6}$$

$$= \int_{\mathbb{R}^n} \boldsymbol{w}_u^T\boldsymbol{x} \Phi(a\boldsymbol{v}_u^T\boldsymbol{x}+b)\mathcal{N}(\boldsymbol{x}|\boldsymbol{\mu}_{u,i},\boldsymbol{\Sigma}_{u,i}) \, d\boldsymbol{x} \tag{7}$$

$$= \int_{\mathbb{R}^n} \boldsymbol{w}_u^T(\boldsymbol{A}_{u,i}^{-1}\boldsymbol{z}+\boldsymbol{\mu}_{u,i}) \Phi(a\boldsymbol{v}_u^T(\boldsymbol{A}_{u,i}^{-1}\boldsymbol{z}+\boldsymbol{\mu}_{u,i})+b)\mathcal{N}(\boldsymbol{z}|\boldsymbol{0},\boldsymbol{I}) \, d\boldsymbol{z} \tag{8}$$

$$= \boldsymbol{w}_u^T\boldsymbol{\mu}_{u,i}\Phi(\frac{a\boldsymbol{v}_u^T\boldsymbol{\mu}_{u,i}+b}{\sqrt{a^2\boldsymbol{v}_u^T\boldsymbol{\Sigma}_{u,i}\boldsymbol{v}_u+1}}) + \frac{a\boldsymbol{v}_u^T\boldsymbol{\Sigma}_{u,i}\boldsymbol{w}_u}{\sqrt{2\pi(a^2\boldsymbol{v}_u^T\boldsymbol{\Sigma}_{u,i}\boldsymbol{v}_u+1)}}e^{-\frac{(a\boldsymbol{v}_u^T\boldsymbol{\mu}_{u,i}+b)^2}{2(a^2\boldsymbol{v}_u^T\boldsymbol{\Sigma}_{u,i}\boldsymbol{v}_u+1)}} \tag{9}$$

Equation 8 and Equation 9 are derived from Proposition 3 and Proposition 5 respectively. Due to page limits, Proposition 3 and Proposition 5 and their detailed proofs are presented in Appendix. Combing Corollary 1 and Equation 9, we have:

$$U_u(i|display;\boldsymbol{\pi}) = \frac{1}{p_{u,i}(display;\boldsymbol{\pi})} \int_{\mathbb{R}^n} f(\boldsymbol{w}_u, \boldsymbol{x}) p_{u,i}(\boldsymbol{x}) p_{u,i}(display|\boldsymbol{x};\boldsymbol{\pi}) \, d\boldsymbol{x} \tag{10}$$

$$= \boldsymbol{w}_u^T\boldsymbol{\mu}_{u,i} + \frac{a\boldsymbol{v}_u^T\boldsymbol{\Sigma}_{u,i}\boldsymbol{w}_u}{\sqrt{2\pi(a^2\boldsymbol{v}_u^T\boldsymbol{\Sigma}_{u,i}\boldsymbol{v}_u+1)}\Phi(\frac{a\boldsymbol{v}_u^T\boldsymbol{\mu}_{u,i}+b}{\sqrt{a^2\boldsymbol{v}_u^T\boldsymbol{\Sigma}_{u,i}\boldsymbol{v}_u+1}})}e^{-\frac{(a\boldsymbol{v}_u^T\boldsymbol{\mu}_{u,i}+b)^2}{2(a^2\boldsymbol{v}_u^T\boldsymbol{\Sigma}_{u,i}\boldsymbol{v}_u+1)}}$$

$$\tag{11}$$

$\Box$

Corollary 2 reveals that the posterior utility can be effectively calculated (since it is with a close-form expression and thus can avoid the intractable calculation in Equation 5) when $\mathbf{\Sigma}_{u,i}$ is positive definite and the utility function $f$ is linear, which makes it possible to be effectively utilized in the real-world scenarios.

However, when the utility function $f$ is non-linear, it might be challenging or even impossible to calculate the exact value of the posterior utility. To estimate the posterior utility when $f$ is non-linear, we leverage the importance sampling technique to approximate the posterior utility. Combing Assumption 2, Equation 5 and Corollary 1, we have:

$$U_u(i|display; \boldsymbol{\pi}) = \int_{\mathbb{R}^n} f(\boldsymbol{w}_u, \boldsymbol{x}) p_{u,i}(\boldsymbol{x}) l(\boldsymbol{x}; \boldsymbol{\pi}) \, d\boldsymbol{x}, \tag{12}$$

where $l(\boldsymbol{x}; \boldsymbol{\pi}) = \frac{\Phi(a\boldsymbol{v}_u^T\boldsymbol{x}+b)}{\Phi(\frac{a\boldsymbol{v}_u^T\boldsymbol{\mu}_{u,i}+b}{\sqrt{a^2\boldsymbol{v}_u^T\mathbf{\Sigma}_{u,i}\boldsymbol{v}_u+1}})}$. Thus, computing $U_u(i|display; \boldsymbol{\pi})$ can be regarded as calcu-

lating the expected value of $f(\boldsymbol{w}_u, X)$ w.r.t. the target probability density function $q(X = \boldsymbol{x}) = p_{u,i}(\boldsymbol{x}) l(\boldsymbol{x}; \boldsymbol{\pi})$. Since $p_{u,i}(\boldsymbol{x})$ is the probability density function of a multivariate Gaussian distribution from which one can easily conduct sampling, we can adopt $p_{u,i}(\boldsymbol{x})$ as the sampler density and as a result, given a sample $\boldsymbol{x}$, the corresponding importance weight is $l(\boldsymbol{x}; \boldsymbol{\pi})$. Thus, given a sample set $\mathbb{X}$ which consists of $m$ samples drawn i.i.d. according to distribution $\mathcal{N}(\boldsymbol{\mu}_{u,i}, \mathbf{\Sigma}_{u,i})$, we can approximate the posterior utility as:

$$\hat{U}_u(i|display; \boldsymbol{\pi}) = \frac{1}{\sum_{\boldsymbol{x} \in \mathbb{X}} l(\boldsymbol{x}; \boldsymbol{\pi})} \sum_{\boldsymbol{x} \in \mathbb{X}} f(\boldsymbol{w}_u, \boldsymbol{x}) l(\boldsymbol{x}; \boldsymbol{\pi}). \tag{13}$$

In this section, we derive an effective calculation formula for posterior utility in the case with linear utility function and provide a sampling-based method to estimate posterior utility in the case with non-linear utility function. Note that the sampling-based method can also be applied to estimate the posterior utility for the case with linear utility function. Combining the results in this section (i.e., Equation 11 and Equation 13) and the formulation of optimal policy (as shown in Equation 1), we notice that the problem of deriving the optimal policy for maximizing platform's expected posterior utility, which is formulated in Equation 1, is turned into a constrained non-convex optimization problem even for the case with linear utility function. To solve the problem, an ADMM-based solution is proposed in the next section.

### 3.3 An ADMM-based Solution for Deriving Approximately Optimal Policy

As presented in the previous section, finding the optimal policy to maximize the expected posterior utility of the platform is formulated as a constrained non-convex optimization problem. Noticing that the ADMM technique (Boyd et al., 2011) has been successfully adopted as an effective method to find approximate solution for constrained non-convex optimization problem (see Leng et al. (2017) for more details), we develop an ADMM-based solution to solve the aforementioned constrained non-convex optimization problem.

Denoting $g_{u,\mathbb{I}}(\boldsymbol{\pi}) = -\sum_{i \in \mathbb{I}} \boldsymbol{\pi}_i U_u(i|display; \boldsymbol{\pi})$, we rewrite the constrained non-convex optimization problem (which is formulated in Equation 1) as follows.

$$\underset{\boldsymbol{\pi}}{argmin} \, g_{u,\mathbb{I}}(\boldsymbol{\pi}), s.t., \forall i \in \mathbb{I}, \boldsymbol{\pi}_i \geq 0 \;\; and \;\; \sum_{i \in \mathbb{I}} \boldsymbol{\pi}_i = 1. \tag{14}$$

By introducing an auxiliary parameter vector $\boldsymbol{\pi}'$, the problem can be reformulated as:

$$\underset{\boldsymbol{\pi}}{argmin} \, g_{u,\mathbb{I}}(\boldsymbol{\pi}'), s.t., \boldsymbol{\pi}' = \boldsymbol{\pi}, \;\; \forall i \in \mathbb{I}, \boldsymbol{\pi}_i \geq 0 \;\; and \;\; \sum_{i \in \mathbb{I}} \boldsymbol{\pi}_i = 1. \tag{15}$$

Note that the target function $g_{u,\mathbb{I}}(\boldsymbol{\pi}')$ and the target policy $\boldsymbol{\pi}$ is linked via the first constrain, i.e., $\boldsymbol{\pi}' = \boldsymbol{\pi}$. We regard the last two constrains as hard constrains which are required to be satisfied at any optimization step. And the augmented Lagrangian w.r.t. to the first constrain is:

$$L_\rho(\boldsymbol{\pi}, \boldsymbol{\pi}', \boldsymbol{\lambda}) = g_{u,\mathbb{I}}(\boldsymbol{\pi}') + \boldsymbol{\lambda}^T(\boldsymbol{\pi}' - \boldsymbol{\pi}) + \frac{\rho}{2}||\boldsymbol{\pi}' - \boldsymbol{\pi}||_2^2, \tag{16}$$

where $\boldsymbol{\lambda}$ is the Lagrangian multipliers and $\rho > 0$ is a hyper-parameter. To achieve an approximate solution, the ADMM method consists of the following three steps for each iteration:

$$\boldsymbol{\pi}^{(k+1)} := \underset{\boldsymbol{\pi}}{argmin}\, L_\rho(\boldsymbol{\pi}, \boldsymbol{\pi}'^{(k)}, \boldsymbol{\lambda}^{(k)}) \tag{17}$$

$$\boldsymbol{\pi}'^{(k+1)} := \underset{\boldsymbol{\pi}'}{argmin}\, L_\rho(\boldsymbol{\pi}^{(k+1)}, \boldsymbol{\pi}', \boldsymbol{\lambda}^{(k)}) \tag{18}$$

$$\boldsymbol{\lambda}^{(k+1)} := \boldsymbol{\lambda}^{(k)} + \rho(\boldsymbol{\pi}'^{(k+1)} - \boldsymbol{\pi}^{(k+1)}) \tag{19}$$

We can observe that the last two steps do not involve optimization w.r.t. $\boldsymbol{\pi}$. Thus, the third step can be simply achieved by value updating while the second step can be approximately solved by adopting gradient-based solutions such as stochastic gradient decent. The optimization problem w.r.t. the first step can be reformulated as:

$$\underset{\boldsymbol{\pi}}{argmin}\, L_\rho(\boldsymbol{\pi}, \boldsymbol{\pi}'^{(k)}, \boldsymbol{\lambda}^{(k)}) = \underset{\boldsymbol{\pi}}{argmin}\, \boldsymbol{\lambda}^{(k)T}(\boldsymbol{\pi}'^{(k)} - \boldsymbol{\pi}) + \frac{\rho}{2}||\boldsymbol{\pi}'^{(k)} - \boldsymbol{\pi}||_2^2 \tag{20}$$

$$= \underset{\boldsymbol{\pi}}{argmin}\, \frac{\rho}{2}||\boldsymbol{\pi}'^{(k)} + \frac{\boldsymbol{\lambda}^{(k)}}{\rho} - \boldsymbol{\pi}||_2^2, \tag{21}$$

$$s.t., \forall i \in \mathbb{I}, \boldsymbol{\pi}_i \geq 0 \ \ and \ \sum_{i \in \mathbb{I}} \boldsymbol{\pi}_i = 1, \tag{22}$$

which is a convex quadratic programming problem and can be solved by convex quadratic programming solutions such as active set methods (Wong, 2011). It is worth noting that the optimization problem w.r.t. the first step can be regarded as finding the nearest point from a given convex set (which satisfy the two constrains) to a given point (i.e., $\boldsymbol{\pi}'^{(k)} + \frac{\boldsymbol{\lambda}^{(k)}}{\rho}$).

By iteratively conducting the aforementioned three optimization steps until convergence (say, after $k'$ steps), we can acquire an approximately optimal policy $\boldsymbol{\pi}^{(k')}$ for maximizing expected posterior utility of the platform, which is demonstrated to be great enough in the experiment section.

## 4 EXPERIMENTS

In this section, we present the experiment setup and the experiment results in detail. First, we present the detailed setup of the experiments including a practical way to gain the mean vector and covariance matrix for given pair of user and item, which makes it possible to take the user uncertainty over different item dimensions into consideration. Second, we conduct experiments to verify the effectiveness of the proposed technical solutions involved in our framework, i.e., the effectiveness of the ADMM-based solution for maximizing platform expected posterior utility and the sampling-based technique for posterior utility estimation. Third, to verify the superior of the policy derived from the proposed framework, we conduct experiments and present the results of comparison between our policy and other two heuristic policies. Fourth, to answer the crucial question that how the platform maximizes its commercial benefits, we analyse the relation between the learned policy and two significant factors, the user preference and uncertainty over given item dimension, and provide practical insights to better understand and conduct commercial recommendation.

### 4.1 EXPERIMENT SETUP

To conduct experiments, we first collect click log data of about 1 million users and 5 million items from the GUESS U LIKE scenario of the Taobao Application. To encode the users and the items, we adopt prevalent Embedding&MLP paradigm (He et al., 2017; Guo et al., 2017) and encode the information of the user and the item sides respectively (Xu et al., 2019). By training on specific task (e.g, click through rate prediction task), we can gain the embedding of the user side as well as that of the item side. Without loss of generality, we consider the following two model architectures: i) the model output is the inner product of the user and the item embeddings; ii) the model output is achieved by applying a non-linear function over the user and the item embeddings. Note that these two architectures correspond to the cases with linear and non-linear utility functions respectively. For each pair of user group (divided according to age and gender) and item category, the clicked item set is extracted and the mean vector and the covariance matrix of the corresponding item embeddings are calculated. For a given pair of user and item, the mean vector and the covariance matrix can be gained by adopting those of the pair of the corresponding user group and item category.

Without loss of generality, we empirically set the value of hyper-parameters $a$ and $b$ to ensure that most of the values of $p_{u,i}(display|\boldsymbol{x}; \boldsymbol{\pi})$ lie between 0.001 and 0.1 (noticing that other reasonable ranges also lead to similar experimental conclusions). Since $\Phi$ is a function in the form of integral which might be intractable to calculate, we adopt an effective approximation (Waissi & Rossin, 1996) to ease the calculation. To avoid learning a highly unbalanced policy (with probability 1 for one item while with 0 for others) due to unbalanced utilities, we also incorporate an entropy-based regularization term into our model. Specifically, instead of directly adopting the optimization step as shown in Equation 18, we add an extra term to smooth the learned policy and the corresponding optimization step is:

$$\boldsymbol{\pi}'^{(k+1)} := \underset{\boldsymbol{\pi}'}{argmin}\, L_\rho(\boldsymbol{\pi}^{(k+1)}, \boldsymbol{\pi}', \boldsymbol{\lambda}^{(k)}) + \eta \sum_{i \in \mathbb{I}}(\boldsymbol{\pi}'_i - \frac{1}{|\mathbb{I}|})^2, \tag{23}$$

where $|\mathbb{I}|$ denotes the number of items in item set $\mathbb{I}$, $\eta$ is the parameter to control the smooth level of the learned policy and is set to 10.0 empirically in the experiments. Besides, the optimizer we adopted for gradient decent is the Adam Optimizer (Kingma & Ba, 2014) and the value of $\rho$ is set as 1.0 empirically, which is proved to be effective in the following experiments.

## 4.2 EFFECTIVENESS OF THE PROPOSED TECHNICAL SOLUTIONS

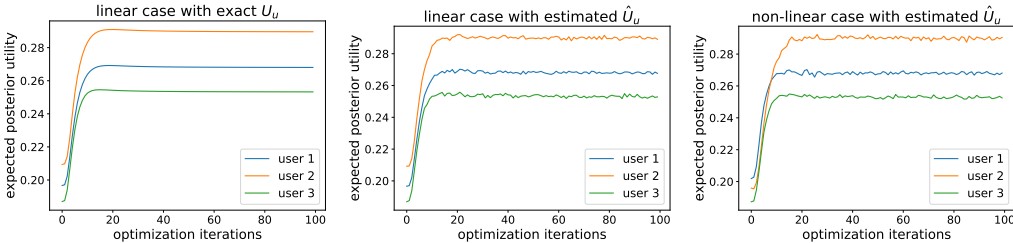

Figure 1: Expected posterior utility w.r.t. optimization iterations for three cases

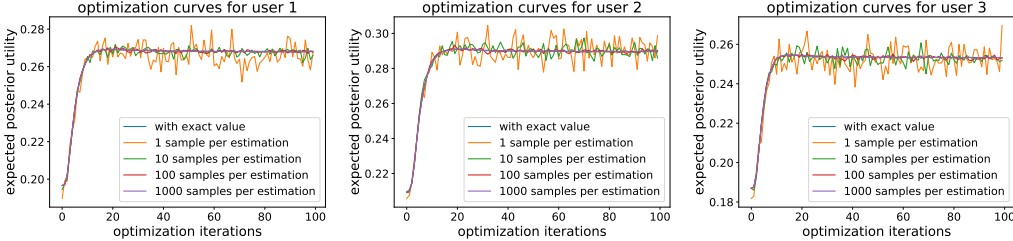

Figure 2: Optimization curves w.r.t. different numbers of samples per estimation

In this section, we present the experiment results to demonstrate the effectiveness of the two proposed technical solutions absorbed into our framework, i.e., the effectiveness of the ADMM-based solution for maximizing expected posterior utility and the importance sampling technique for posterior utility estimation.

To demonstrate the effectiveness of the proposed ADMM-based solution for maximizing expected posterior utility, we record the optimization curves (i.e., expected posterior utility w.r.t. the number of optimization iterations) and present the results in Figure 1. According to Section 3.2, there are three cases to be considered: i) linear utility function with exact posterior utility (calculated by adopting Equation 11); ii) linear utility function with estimated posterior utility; iii) non-linear utility function with estimated posterior utility. We randomly sample 3 users and 100 items and utilize the proposed ADMM-based optimization method to learn the approximately optimal policy to maximize the expected posterior utility of the platform. As shown in Figure 1, the expected posterior utility first increases and then converges to its maximum with the increase of the number of optimization iterations. These results demonstrate that the proposed ADMM-based solution can effectively learn a policy to maximize the expected posterior utility. Besides, comparing the optimization curves of the three cases, we can observe that the variances of the expected posterior utility in the cases with

estimated posterior utility $\hat{U}_u(i|display; \boldsymbol{\pi})$ (the last two cases) are higher than that in the case with exact calculated posterior utility $U_u(i|display; \boldsymbol{\pi})$ (the first case). The reason is that extra random noise is induced by importance sampling when we conduct posterior utility estimation and as a result, expected posterior utility is with higher variance in the case with estimated posterior utility.

In order to verify the effectiveness of the adopted importance sampling technique for posterior estimation, we record the optimization curves w.r.t. different numbers of samples for estimating posterior utility for three randomly sampled users and present the results in Figure 2. The linear utility function is adopted to conduct the experiments which makes it possible to make comparison between the results of utilizing estimated posterior utility (estimated according to Equation 13) and exact posterior utility (calculated according to Equation 11). From Figure 2 we can observe that even in the case with 1 sample per estimation, the tendency of the optimization curve is similar to that of the case with exact posterior utility, which verifies that the adopted sampling-based technique for posterior utility estimation is effective. Besides, when varying the number of samples per estimation from 1 to 1000, we observe that the optimization curves become more and more stable. It is because that larger number of samples per estimation leads to more accurate approximation of posterior utility, and thus results in more stable optimization curve. More details including the Root Mean Square Error (RMSE) between the estimated values and the exact values of the posterior utilities for each sampling setting are provided in Appendix A.4 due to page limits.

## 4.3 EXPERIMENTAL COMPARISON WITH HEURISTIC POLICIES

In this section, we present the experiment details and the corresponding results to verify the effectiveness of the policy derived from the proposed framework. To the best of our knowledge, the presented framework is the first complete work that proposes to maximize the expected posterior utility (which explicitly takes the impact of display policy into consideration) and develops a corresponding solution to derive the approximately optimal policy. Thus, there may lack solutions for conducting straightforward comparison and we intent to verify the effectiveness of the learned policy by comparing it with other heuristic policies. One simple policy for comparison is the random policy that recommends each item with same probability. Thus, the random policy can be regarded as an uniform probability distribution over the candidate items. We also incorporate a heuristic prior policy for comparison where the recommendation probability is higher for item with higher prior utility[2]. Specifically, the recommendation probability of item $i$ is given by

$$\boldsymbol{\pi}_i^{pop} = \frac{e^{\frac{U_u(i)}{\tau}}}{\sum_{j \in \mathbb{I}} e^{\frac{U_u(j)}{\tau}}} \tag{24}$$

where $\tau$ is the temperature parameter (Hinton et al., 2015) to control the smooth level of the heuristic prior policy. For conducting fair comparison, we adjust the value of $\tau$ to ensure that the smooth levels of the heuristic prior policy and our learned policy are comparable.

Table 1: Expected posterior utilities w.r.t. different policies

|  | random policy | prior policy | our policy |
|---|---|---|---|
| linear case | 0.1593 | 0.1922 | **0.2171** |
| non-linear case | 0.1609 | 0.1936 | **0.2156** |

We randomly sample 1000 users, calculate the mean of expected posterior utilities for each policy and present the results in Table 1. As shown in Table 1, the policy derived from the proposed framework achieves the highest expected posterior utility compared to the random and the heuristic prior policies in both linear and non-linear cases, which demonstrates the superior of the proposed framework.

## 4.4 HOW DOES THE PLATFORM ACHIEVE ITS COMMERCIAL BENEFITS

In this section, we intent to adopt the proposed framework to reveal how the platform achieves its commercial benefits taking both the user preference and the user uncertainty over different item dimensions into account. In the experiments, the utility function is realized by applying the sigmoid

---

[2]Similar to the derivation of posterior utility, we can derive that i) for linear case, $U_u(i) = \boldsymbol{w}_u^T \boldsymbol{\mu}_{u,i}$; ii) for non-linear case, $U_u(i) = \int_{\mathbb{R}^n} f(\boldsymbol{w}_u, \boldsymbol{x}) p_{u,i}(\boldsymbol{x}) \, d\boldsymbol{x}$.

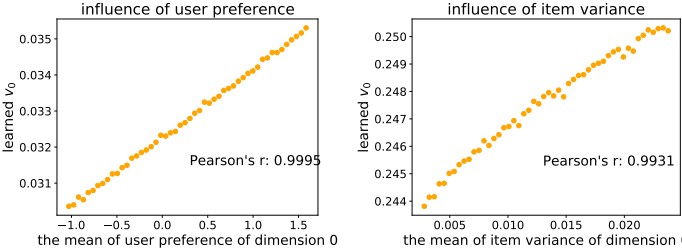

Figure 3: Influence of user preference and item variance

activation function over the inner product of item representation and user representation (i.e., $\boldsymbol{w}_u$). Each element of $\boldsymbol{w}_u$ can be explained as the user preference over the corresponding item dimension. For a randomly sampled user, we denote the mean vector of the user preference vectors over items as $\bar{\boldsymbol{w}}$ and denote its first element as $\bar{\boldsymbol{w}}_0$, whose value indicates the user's preference of item dimension 0. We vary the value of $\bar{\boldsymbol{w}}_0$ and record the learned expected value (w.r.t. the learned policy) of the elements with dimension 0 of the mean vectors, which is denoted as $\boldsymbol{v}_0$. As shown in the left part of Figure 3, with the increase of $\bar{\boldsymbol{w}}_0$, the value of $\boldsymbol{v}_0$ increases approximately linearly (noting that the Pearson correlation coefficient is 0.9995 which indicates strong linear correlation). According to its definition, $\boldsymbol{v}_0$ can reflect to what extent the policy has adjusted to fit item dimension 0 (more details about how the policy adjust to different levels of user preference or uncertainty are provided in Appendix A.5). Thus, the result suggests that the policy would tend to adjust more for dimensions with higher value. In other words, the platform can achieve more commercial benefits from the item dimension with stronger user preference. We also vary the mean of item variance of dimension 0, which indicates the user uncertainty over item dimension 0, and record $\boldsymbol{v}_0$ as presented in the right part of Figure 3. By similar analysis, the result indicates that the platform can achieve more commercial benefits from the item dimension that the user is with higher uncertainty.

Therefore, the results in Figure 3 indicate that commercial benefits mainly come from the item dimensions with either strong user preference or high user uncertainty when we take the impact of display policy on user into consideration. Note that this result may contribute to better understand how commercial benefits are achieved and better conduct commercial recommendation. For instance, from the perspective of the platform, one effective way to improve its commercial benefits is to better balance the user experience and its commercial actions, i.e., for item dimensions where the user is with high preference, the platform should cater to his preference, while for flexible item dimensions where the user is with high uncertainty, the platform may achieve more benefits by recommending items with high platform utilities.

## 5 CONCLUSION

In the paper, we propose a novel recommendation framework and take into account the user uncertainty over different item dimensions and the influence of display policy over user, both of which could highly affect the commercial benefits of the platform. We derive the calculation formula of the posterior utility in the case with linear utility function and provide a sampling-based method to estimate the posterior utility in the case with non-linear utility function. Based on these works, we formulate the problem of deriving optimal policy for maximizing the expected posterior utility of the platform as a constrained non-convex optimization problem and further propose an ADMM-based solution to derive an approximately optimal policy. To demonstrate the effectiveness of the proposed technical solutions absorbed into our framework, extensive experiments are conducted over data collected from a real-world recommendation scenario. Furthermore, we also provide some practical insights about how to achieve commercial benefits for the platform taking both the user preference and the user uncertainty over different item dimensions into consideration, which might contribute to better understand and conduct commercial recommendation.

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

## A APPENDIX

### A.1 PROOF OF LEMMA 1

To prove Lemma 1, we first prove the following two propositions.

**Proposition 1.** $\int_{-\infty}^{+\infty} \Phi(a'x + b')\mathcal{N}(x|\mu_0, \sigma_0^2)\, dx = \Phi(\frac{a'\mu_0 + b'}{\sqrt{1+a'^2\sigma_0^2}}), \forall a', b', \mu_0, \sigma_0 \in \mathbb{R}$.

*Proof.* By introducing two independent variables $X$ and $Y$ which satisfy $X \sim \mathcal{N}(\mu_0, \sigma_0)$ and $Y \sim \mathcal{N}(0, 1)$, we have

$$\int_{-\infty}^{+\infty} \Phi(a'x + b')\mathcal{N}(x|\mu_0, \sigma_0^2)\, dx = E_{X \sim \mathcal{N}(\mu_0, \sigma_0)}(\Phi(a'X + b')) \tag{25}$$

$$= \mathbb{P}(Y < a'X + b') \tag{26}$$

$$= \mathbb{P}(Y - a'X < b') \tag{27}$$

$$= \mathbb{P}(\frac{Y - a'X + a'\mu_0}{\sqrt{1 + a'^2\sigma_0^2}} < \frac{b' + a'\mu_0}{\sqrt{1 + a'^2\sigma_0^2}}) \tag{28}$$

$$= \Phi(\frac{a'\mu_0 + b'}{\sqrt{1 + a'^2\sigma_0^2}}) \tag{29}$$

Note that Equation 26 is gained by the definition of function $\Phi$, i.e., $\Phi(z)$ is the probability of sampling from the standard normal distribution with the sampled value less than $z$. Since $X \sim \mathcal{N}(\mu_0, \sigma_0)$ and $Y \sim \mathcal{N}(0, 1)$, we have $\frac{Y - a'X + a'\mu_0}{\sqrt{1+a'^2\sigma_0^2}} \sim \mathcal{N}(0, 1)$. Thus, Equation 29 can be straightforwardly derived by definition of $\Phi$, which concludes the proof. $\square$

**Proposition 2.** *Denoting the error function by $erf$ which satisfies $erf(z) = \frac{2}{\sqrt{\pi}} \int_0^z e^{-t^2}\, dt$, we have:* $\int_{-\infty}^{\infty} xe^{\frac{-x^2}{2}} erf(a'x + b')\, dx = \frac{2a'}{\sqrt{a'^2 + \frac{1}{2}}} e^{-\frac{b'^2}{2a'^2+1}}, \forall a', b' \in \mathbb{R}$.

*Proof.* For case with $a' = 0$, we have:

$$\int_{-\infty}^{\infty} xe^{\frac{-x^2}{2}} erf(a'x + b')\, dx = \int_{-\infty}^{0} xe^{\frac{-x^2}{2}} erf(b')\, dx + \int_{0}^{\infty} xe^{\frac{-x^2}{2}} erf(b')\, dx \tag{30}$$

$$= -\int_{0}^{\infty} xe^{\frac{-x^2}{2}} erf(b')\, dx + \int_{0}^{\infty} xe^{\frac{-x^2}{2}} erf(b')\, dx \tag{31}$$

$$= 0 \tag{32}$$

$$= \frac{2a'}{\sqrt{a'^2 + \frac{1}{2}}} e^{-\frac{b'^2}{2a'^2+1}} \tag{33}$$

For case with $a' \neq 0$, we have:

$$\int_{-\infty}^{\infty} x e^{\frac{-x^2}{2}} erf(a'x + b') \, dx = -e^{\frac{-x^2}{2}} erf(a'x + b') \Big|_{-\infty}^{\infty} - \int_{-\infty}^{\infty} -e^{\frac{-x^2}{2}} \frac{2a'}{\sqrt{\pi}} e^{-(a'x+b')^2} \, dx \tag{34}$$

$$= \int_{-\infty}^{\infty} \frac{2a'}{\sqrt{\pi}} e^{-(a'^2 + \frac{1}{2})x^2 - 2a'b'x - b'^2} \, dx \tag{35}$$

$$= \int_{-\infty}^{\infty} \frac{2a'}{\sqrt{\pi}} e^{-(a'^2 + \frac{1}{2})(x + \frac{a'b'}{a'^2 + \frac{1}{2}})^2 + \frac{a'^2 b'^2}{a'^2 + \frac{1}{2}} - b'^2} \, dx \tag{36}$$

$$= \int_{-\infty}^{\infty} \frac{2a'}{\sqrt{\pi}} e^{\frac{a'^2 b'^2}{a'^2 + \frac{1}{2}} - b'^2} e^{-(a'^2 + \frac{1}{2})z^2} \, dz \tag{37}$$

$$= \frac{2a'}{\sqrt{\pi}} e^{\frac{a'^2 b'^2}{a'^2 + \frac{1}{2}} - b'^2} \sqrt{\frac{\pi}{a'^2 + \frac{1}{2}}} \tag{38}$$

$$= \frac{2a'}{\sqrt{a'^2 + \frac{1}{2}}} e^{-\frac{b'^2}{2a'^2 + 1}} \tag{39}$$

Note that Equation 34 is derived by applying Newton-Leibniz formula.

Thus, combining the above two cases, we have: $\int_{-\infty}^{\infty} x e^{\frac{-x^2}{2}} erf(a'x + b') \, dx = \frac{2a'}{\sqrt{a'^2 + \frac{1}{2}}} e^{-\frac{b'^2}{2a'^2 + 1}}, \forall a', b' \in \mathbb{R}$. $\qquad\square$

With Proposition 1 and Proposition 2, now we prove Lemma 1 as follows.

*Proof of Lemma 1.* By definition, $\Phi(z) = \frac{1 + erf(\frac{z}{\sqrt{2}})}{2}$. Thus, we have:

$$\int_{-\infty}^{+\infty} (c'x + d')\Phi(a'x + b')\mathcal{N}(x|0, 1) \, dx \tag{40}$$

$$= \int_{-\infty}^{+\infty} \frac{1}{2\sqrt{2\pi}} (c'x)(1 + erf(\frac{a'x + b'}{\sqrt{2}})) e^{-\frac{x^2}{2}} \, dx + \int_{-\infty}^{+\infty} d'\Phi(a'x + b')\mathcal{N}(x|0, 1) \, dx \tag{41}$$

$$= \frac{a'c'}{\sqrt{2\pi(a'^2 + 1)}} e^{-\frac{b'^2}{2a'^2 + 2}} + d'\Phi(\frac{b'}{\sqrt{1 + a'^2}}) \tag{42}$$

The last equation is derived by utilizing Proposition 1 and Proposition 2, which completes the proof. $\qquad\square$

## A.2 PROOF OF COROLLARY 1

In order to prove Corollary 1, we first prove the following two propositions.

**Proposition 3.** *If $\Sigma_{u,i}$ is positive definite, then $\int_{\mathbb{R}^n} h(\boldsymbol{x})\mathcal{N}(\boldsymbol{x}|\boldsymbol{\mu}_{u,i}, \Sigma_{u,i}) \, d\boldsymbol{x} = \int_{\mathbb{R}^n} h(\boldsymbol{A}_{u,i}^{-1}\boldsymbol{z} + \boldsymbol{\mu}_{u,i})\mathcal{N}(\boldsymbol{z}|\boldsymbol{0}, \boldsymbol{I}) \, d\boldsymbol{z}$, where $\boldsymbol{A}_{u,i}$ is the upper triangular matrix gained by Cholesky decomposition over $\Sigma_{u,i}^{-1}$, i.e., $\Sigma_{u,i}^{-1} = \boldsymbol{A}_{u,i}^T \boldsymbol{A}_{u,i}$*

*Proof.* If $\Sigma_{u,i}$ is positive definite, $\Sigma_{u,i}^{-1}$ is also positive definite. Then we can conduct Cholesky decomposition:

$$\Sigma_{u,i}^{-1} = \boldsymbol{A}_{u,i}^T \boldsymbol{A}_{u,i}, \tag{43}$$

where $\boldsymbol{A}_{u,i}$ is an upper triangular matrix with real and positive diagonal entries.

Thus, by adopting integration by substitution, we have:

$$\int_{\mathbb{R}^n} h(\boldsymbol{x})\mathcal{N}(\boldsymbol{x}|\boldsymbol{\mu}_{u,i}, \Sigma_{u,i}) \, d\boldsymbol{x} \tag{44}$$

$$= \int_{\mathbb{R}^n} h(\boldsymbol{x}) \frac{1}{\sqrt{(2\pi)^n |\boldsymbol{\Sigma}_{u,i}|}} e^{-\frac{1}{2}(\boldsymbol{x}-\boldsymbol{\mu}_{u,i})^T \boldsymbol{\Sigma}_{u,i}^{-1}(\boldsymbol{x}-\boldsymbol{\mu}_{u,i})} \, d\boldsymbol{x} \tag{45}$$

$$= \int_{\mathbb{R}^n} h(\boldsymbol{A}_{u,i}^{-1}\boldsymbol{A}_{u,i}(\boldsymbol{x}-\boldsymbol{\mu}_{u,i}) + \boldsymbol{\mu}_{u,i}) \frac{1}{\sqrt{(2\pi)^n}} e^{-\frac{1}{2}(\boldsymbol{A}_{u,i}(\boldsymbol{x}-\boldsymbol{\mu}_{u,i}))^T \boldsymbol{A}_{u,i}(\boldsymbol{x}-\boldsymbol{\mu}_{u,i})} |\boldsymbol{A}_{u,i}| \, d\boldsymbol{x} \tag{46}$$

$$= \int_{\mathbb{R}^n} h(\boldsymbol{A}_{u,i}^{-1}\boldsymbol{z} + \boldsymbol{\mu}_{u,i}) \frac{1}{\sqrt{(2\pi)^n}} e^{-\frac{1}{2}\boldsymbol{z}^T \boldsymbol{z}} \, d\boldsymbol{z} \tag{47}$$

$$= \int_{\mathbb{R}^n} h(\boldsymbol{A}_{u,i}^{-1}\boldsymbol{z} + \boldsymbol{\mu}_{u,i}) \mathcal{N}(\boldsymbol{z}|\boldsymbol{0}, \boldsymbol{I}) \, d\boldsymbol{z} \tag{48}$$

Thus, if $\boldsymbol{\Sigma}_{u,i}$ is positive definite and $\boldsymbol{A}_{u,i}$ is the upper triangular matrix gained by Cholesky decomposition over $\boldsymbol{\Sigma}_{u,i}^{-1}$, then we have $\int_{\mathbb{R}^n} h(\boldsymbol{x})\mathcal{N}(\boldsymbol{x}|\boldsymbol{\mu}_{u,i}, \boldsymbol{\Sigma}_{u,i}) \, d\boldsymbol{x} = \int_{\mathbb{R}^n} h(\boldsymbol{A}_{u,i}^{-1}\boldsymbol{z} + \boldsymbol{\mu}_{u,i})\mathcal{N}(\boldsymbol{z}|\boldsymbol{0}, \boldsymbol{I}) \, d\boldsymbol{z}$, which concludes the proof. $\square$

**Proposition 4.** $\int_{\mathbb{R}^n} \Phi(\boldsymbol{v}^T \boldsymbol{x} + a') \mathcal{N}(\boldsymbol{x}|\boldsymbol{0}, \boldsymbol{I}) \, d\boldsymbol{x} = \Phi(\frac{a'}{\sqrt{1+\boldsymbol{v}^T \boldsymbol{v}}})$, $\forall \boldsymbol{v} \in \mathbb{R}^n$ and $a' \in \mathbb{R}$.

*Proof.* Denoting $\tilde{\boldsymbol{v}}_i = a' + \sum_{j=i+1}^{n} \boldsymbol{v}_j \boldsymbol{x}_j$, we have:

$$\int_{\mathbb{R}^n} \Phi(\boldsymbol{v}^T \boldsymbol{x} + a') \mathcal{N}(\boldsymbol{x}|\boldsymbol{0}, \boldsymbol{I}) \, d\boldsymbol{x} \tag{49}$$

$$= \int_{-\infty}^{+\infty} \cdots \int_{-\infty}^{+\infty} \Phi(a' + \sum_{j=1}^{n} \boldsymbol{v}_j \boldsymbol{x}_j) \mathcal{N}(\boldsymbol{x}_1|0, 1) \, d\boldsymbol{x}_1 \cdots \mathcal{N}(\boldsymbol{x}_n|0, 1) \, d\boldsymbol{x}_n \tag{50}$$

$$= \int_{-\infty}^{+\infty} \cdots \int_{-\infty}^{+\infty} \Phi(\boldsymbol{v}_1 \boldsymbol{x}_1 + \tilde{\boldsymbol{v}}_1) \mathcal{N}(\boldsymbol{x}_1|0, 1) \, d\boldsymbol{x}_1 \cdots \mathcal{N}(\boldsymbol{x}_n|0, 1) \, d\boldsymbol{x}_n \tag{51}$$

$$= \int_{-\infty}^{+\infty} \cdots \int_{-\infty}^{+\infty} \Phi(\frac{\tilde{\boldsymbol{v}}_1}{\sqrt{1 + \boldsymbol{v}_1^2}}) \mathcal{N}(\boldsymbol{x}_2|0, 1) \, d\boldsymbol{x}_2 \cdots \mathcal{N}(\boldsymbol{x}_n|0, 1) \, d\boldsymbol{x}_n \tag{52}$$

$$= \int_{-\infty}^{+\infty} \cdots \int_{-\infty}^{+\infty} \Phi(\frac{\boldsymbol{v}_2 \boldsymbol{x}_2 + \tilde{\boldsymbol{v}}_2}{\sqrt{1 + \boldsymbol{v}_1^2}}) \mathcal{N}(\boldsymbol{x}_2|0, 1) \, d\boldsymbol{x}_2 \cdots \mathcal{N}(\boldsymbol{x}_n|0, 1) \, d\boldsymbol{x}_n \tag{53}$$

$$= \int_{-\infty}^{+\infty} \cdots \int_{-\infty}^{+\infty} \Phi(\frac{\frac{\tilde{\boldsymbol{v}}_2}{\sqrt{1+\boldsymbol{v}_1^2}}}{\sqrt{1 + \frac{\boldsymbol{v}_2^2}{1+\boldsymbol{v}_1^2}}}) \mathcal{N}(\boldsymbol{x}_3|0, 1) \, d\boldsymbol{x}_3 \cdots \mathcal{N}(\boldsymbol{x}_n|0, 1) \, d\boldsymbol{x}_n \tag{54}$$

$$= \int_{-\infty}^{+\infty} \cdots \int_{-\infty}^{+\infty} \Phi(\frac{\tilde{\boldsymbol{v}}_2}{\sqrt{1 + \boldsymbol{v}_1^2 + \boldsymbol{v}_2^2}}) \mathcal{N}(\boldsymbol{x}_3|0, 1) \, d\boldsymbol{x}_3 \cdots \mathcal{N}(\boldsymbol{x}_n|0, 1) \, d\boldsymbol{x}_n \tag{55}$$

$$= \Phi(\frac{a'}{\sqrt{1 + \boldsymbol{v}^T \boldsymbol{v}}}) \tag{56}$$

Equation 52 and Equation 54 are derived from Lemma 1 (or Proposition 1). Similarly, Equation 56 is achieved by applying Lemma 1 (or Proposition 1) iteratively, which completes the proof. $\square$

With Proposition 3 and Proposition 4, now we can prove Corollary 1 as follows.

*Proof of Corollary 1.* If $\boldsymbol{\Sigma}_{u,i}$ is positive definite, $\boldsymbol{\Sigma}_{u,i}^{-1}$ is also positive definite. Then we can conduct Cholesky decomposition: $\boldsymbol{\Sigma}_{u,i}^{-1} = \boldsymbol{A}_{u,i}^T \boldsymbol{A}_{u,i}$, where $\boldsymbol{A}_{u,i}$ is an upper triangular matrix with real and positive diagonal entries. Thus, we have:

$$p_{u,i}(display; \boldsymbol{\pi}) = \int_{\mathbb{R}^n} p_{u,i}(display|\boldsymbol{x}; \boldsymbol{\pi}) p_{u,i}(\boldsymbol{x}) \, d\boldsymbol{x} \tag{57}$$

$$= \int_{\mathbb{R}^n} \Phi(a(\boldsymbol{v}_u^T \boldsymbol{x}) + b) \mathcal{N}(\boldsymbol{x}|\boldsymbol{\mu}_{u,i}, \boldsymbol{\Sigma}_{u,i}) \, d\boldsymbol{x} \tag{58}$$

$$= \int_{\mathbb{R}^n} \Phi(a\boldsymbol{v}_u^T \boldsymbol{A}_{u,i}^{-1}\boldsymbol{z} + a\boldsymbol{v}_u^T \boldsymbol{\mu}_{u,i} + b) \mathcal{N}(\boldsymbol{z}|\boldsymbol{0}, \boldsymbol{I}) \, d\boldsymbol{z} \tag{59}$$

$$= \Phi(\frac{a\boldsymbol{v}_u^T\boldsymbol{\mu}_{u,i} + b}{\sqrt{1 + a^2\boldsymbol{v}_u^T\boldsymbol{\Sigma}_{u,i}\boldsymbol{v}_u}}) \tag{60}$$

Equation 59 and Equation 60 are derived from Proposition 3 and Proposition 4 respectively, which concludes the proof. □

### A.3 PROPOSITION ADOPTED FOR PROVING COROLLARY 2

**Proposition 5.** $\int_{\mathbb{R}^n} (\boldsymbol{u}^T\boldsymbol{x} + b')\Phi(\boldsymbol{v}^T\boldsymbol{x} + a')\mathcal{N}(\boldsymbol{x}|\boldsymbol{0},\boldsymbol{I})\, d\boldsymbol{x} = \frac{\boldsymbol{v}^T\boldsymbol{u}}{\sqrt{2\pi(\boldsymbol{v}^T\boldsymbol{v}+1)}}e^{-\frac{a'^2}{2(\boldsymbol{v}^T\boldsymbol{v}+1)}} + b'\Phi(\frac{a'}{\sqrt{\boldsymbol{v}^T\boldsymbol{v}+1}}), \forall \boldsymbol{u}, \boldsymbol{v} \in \mathbb{R}^n \text{ and } a', b' \in \mathbb{R}$.

*Proof.* Denoting $\tilde{\boldsymbol{u}}_i = b' + \sum_{j=i+1}^n \boldsymbol{u}_j\boldsymbol{x}_j$ and $\tilde{\boldsymbol{v}}_i = a' + \sum_{j=i+1}^n \boldsymbol{v}_j\boldsymbol{x}_j$, we have:

$$\int_{\mathbb{R}^n}(\boldsymbol{u}^T\boldsymbol{x} + b')\Phi(\boldsymbol{v}^T\boldsymbol{x} + a')\mathcal{N}(\boldsymbol{x}|\boldsymbol{0},\boldsymbol{I})\, d\boldsymbol{x} \tag{61}$$

$$= \int_{-\infty}^{+\infty}\cdots\int_{-\infty}^{+\infty}(b' + \sum_{j=1}^n \boldsymbol{u}_j\boldsymbol{x}_j)\Phi(a' + \sum_{j=1}^n \boldsymbol{v}_j\boldsymbol{x}_j)\mathcal{N}(\boldsymbol{x}_1|0,1)\, d\boldsymbol{x}_1\cdots\mathcal{N}(\boldsymbol{x}_n|0,1)\, d\boldsymbol{x}_n \tag{62}$$

$$= \int_{-\infty}^{+\infty}\cdots\int_{-\infty}^{+\infty}(\boldsymbol{u}_1\boldsymbol{x}_1 + \tilde{\boldsymbol{u}}_1)\Phi(\boldsymbol{v}_1\boldsymbol{x}_1 + \tilde{\boldsymbol{v}}_1)\mathcal{N}(\boldsymbol{x}_1|0,1)\, d\boldsymbol{x}_1\cdots\mathcal{N}(\boldsymbol{x}_n|0,1)\, d\boldsymbol{x}_n \tag{63}$$

$$= \int_{-\infty}^{+\infty}\cdots\int_{-\infty}^{+\infty}(\frac{\boldsymbol{v}_1\boldsymbol{u}_1}{\sqrt{2\pi(\boldsymbol{v}_1^2+1)}}e^{-\frac{\tilde{\boldsymbol{v}}_1^2}{2(\boldsymbol{v}_1^2+1)}} + \tilde{\boldsymbol{u}}_1\Phi(\frac{\tilde{\boldsymbol{v}}_1}{\sqrt{1+\boldsymbol{v}_1^2}}))\mathcal{N}(\boldsymbol{x}_2|0,1)\, d\boldsymbol{x}_2\cdots\mathcal{N}(\boldsymbol{x}_n|0,1)\, d\boldsymbol{x}_n \tag{64}$$

$$= \int_{-\infty}^{+\infty}\cdots\int_{-\infty}^{+\infty}(\frac{\boldsymbol{v}_1\boldsymbol{u}_1}{\sqrt{2\pi(\boldsymbol{v}_1^2+1)}}e^{-\frac{(\boldsymbol{v}_2\boldsymbol{x}_2+\tilde{\boldsymbol{v}}_2)^2}{2(\boldsymbol{v}_1^2+1)}} + (\boldsymbol{u}_2\boldsymbol{x}_2 + \tilde{\boldsymbol{u}}_2)\Phi(\frac{\boldsymbol{v}_2\boldsymbol{x}_2 + \tilde{\boldsymbol{v}}_2}{\sqrt{1+\boldsymbol{v}_1^2}}))\mathcal{N}(\boldsymbol{x}_2|0,1)\, d\boldsymbol{x}_2$$
$$\cdots\mathcal{N}(\boldsymbol{x}_n|0,1)\, d\boldsymbol{x}_n \tag{65}$$

$$= \int_{-\infty}^{+\infty}\cdots\int_{-\infty}^{+\infty}(\frac{\boldsymbol{v}_1\boldsymbol{u}_1}{\sqrt{2\pi(\boldsymbol{v}_1^2+\boldsymbol{v}_2^2+1)}}e^{-\frac{(\boldsymbol{v}_3\boldsymbol{x}_3+\tilde{\boldsymbol{v}}_3)^2}{2(\boldsymbol{v}_1^2+\boldsymbol{v}_2^2+1)}} + \frac{\boldsymbol{v}_2\boldsymbol{u}_2}{\sqrt{2\pi(\boldsymbol{v}_1^2+\boldsymbol{v}_2^2+1)}}e^{-\frac{(\boldsymbol{v}_3\boldsymbol{x}_3+\tilde{\boldsymbol{v}}_3)^2}{2(\boldsymbol{v}_1^2+\boldsymbol{v}_2^2+1)}} +$$
$$(\boldsymbol{u}_3\boldsymbol{x}_3 + \tilde{\boldsymbol{u}}_3)\Phi(\frac{\boldsymbol{v}_3\boldsymbol{x}_3 + \tilde{\boldsymbol{v}}_3}{\sqrt{1+\boldsymbol{v}_1^2+\boldsymbol{v}_2^2}}))\mathcal{N}(\boldsymbol{x}_3|0,1)\, d\boldsymbol{x}_3\cdots\mathcal{N}(\boldsymbol{x}_n|0,1)\, d\boldsymbol{x}_n \tag{66}$$

$$= \int_{-\infty}^{+\infty}\cdots\int_{-\infty}^{+\infty}(\frac{\boldsymbol{v}_1\boldsymbol{u}_1 + \boldsymbol{v}_2\boldsymbol{u}_2}{\sqrt{2\pi(\boldsymbol{v}_1^2+\boldsymbol{v}_2^2+1)}}e^{-\frac{(\boldsymbol{v}_3\boldsymbol{x}_3+\tilde{\boldsymbol{v}}_3)^2}{2(\boldsymbol{v}_1^2+\boldsymbol{v}_2^2+1)}} + (\boldsymbol{u}_3\boldsymbol{x}_3 + \tilde{\boldsymbol{u}}_3)\Phi(\frac{\boldsymbol{v}_3\boldsymbol{x}_3 + \tilde{\boldsymbol{v}}_3}{\sqrt{1+\boldsymbol{v}_1^2+\boldsymbol{v}_2^2}}))$$
$$\mathcal{N}(\boldsymbol{x}_3|0,1)\, d\boldsymbol{x}_3\cdots\mathcal{N}(\boldsymbol{x}_n|0,1)\, d\boldsymbol{x}_n \tag{67}$$

$$= \frac{\boldsymbol{v}^T\boldsymbol{u}}{\sqrt{2\pi(\boldsymbol{v}^T\boldsymbol{v}+1)}}e^{-\frac{a'^2}{2(\boldsymbol{v}^T\boldsymbol{v}+1)}} + b'\Phi(\frac{a'}{\sqrt{\boldsymbol{v}^T\boldsymbol{v}+1}}) \tag{68}$$

Equation 64 is derived from Lemma 1. Applying Lemma 1 and conducting definite integration over the exponential function, we can derive Equation 66. Similar calculation step can be conducted iteratively to derive Equation 68, which completes the proof. □

### A.4 ADDITIONAL EXPERIMENTS TO VERIFY THE EFFECTIVENESS OF THE IMPORTANCE SAMPLING BASED POSTERIOR UTILITY ESTIMATION

Table 2: Statistics of posterior utilities w.r.t. samples per estimation

| | exact value | samples per estimation | | | |
|---|---|---|---|---|---|
| | | 1 | 10 | 100 | 1000 |
| RMSE of posterior utilities | 0.0000 | 0.0369 | 0.0125 | 0.0038 | 0.0011 |
| mean of expected posterior utilities | 0.2681 | 0.2660 | 0.2677 | 0.2681 | 0.2681 |
| variance of expected posterior utilities | 3.4917e-9 | 3.5331e-5 | 3.0590e-6 | 2.9973e-7 | 4.3271e-8 |

As described in Section 3.2, importance sampling technique can be adopted to estimate the value of posterior utility. However, it is apparent that the number of samples for each posterior utility estimation would highly affect the results. Extra experiments are conducted to verify the effectiveness of the importance sampling based posterior utility estimation as shown in Table 2. When we vary the number of samples per estimation from 1 to 1000, the Root Mean Square Error (RMSE) between the estimated values and the exact values of the posterior utilities decreases from 0.0369 to 0.0011, which indicates that larger number of samples per estimation leads to more accurate approximation. We also utilize the proposed ADMM-based solution to maximize the expected posterior utility and record the mean and variance of the expected posterior utilities of the last 50 optimization iterations for each sampling setting. For the case with the number of samples per estimation set to 100 (or 1000), the variance of the expected posterior utilities is very small while the learned final value of expected posterior utility is around 0.2681, which is similar to the result of optimization with exact value of posterior utility and thus verifies the effectiveness of the proposed importance sampling based posterior utility estimation.

## A.5 ADDITIONAL EXPERIMENTS TO ILLUSTRATE HOW THE POLICY ADJUST W.R.T. DIFFERENT LEVELS OF USER PREFERENCE OR UNCERTAINTY

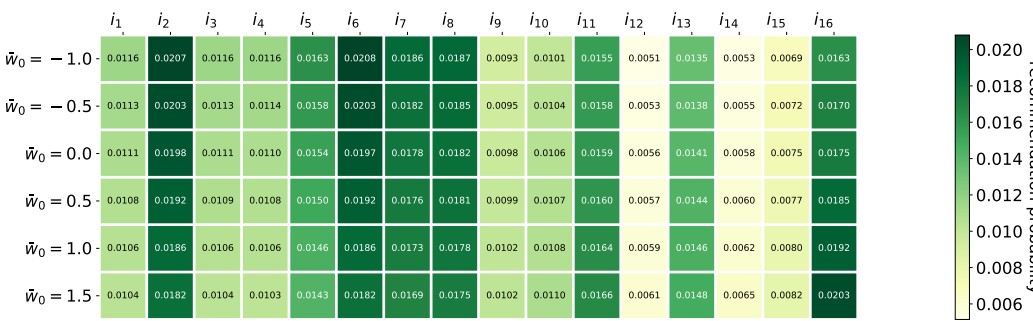

Figure 4: Learned policy w.r.t. different levels of user preference

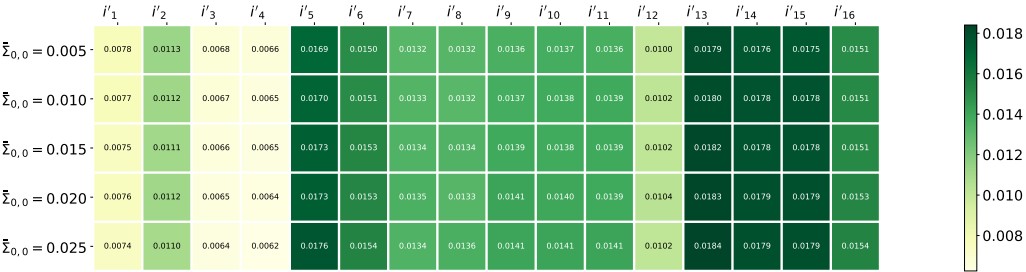

Figure 5: Learned policy w.r.t. different levels of user uncertainty

We also provide analysis about how different levels of user preference would affect the final learned policy over items. For a randomly sampled user, we vary $\bar{w}_0$ from -1.0 to 1.5 and keep the values of other dimensions unchanged. We also randomly pick 16 items from candidate item set, sort these items according to the value of dimension 0 of the item mean vector and denote the sorted items as $i_1, i_2, \cdots, i_{16}$. For each case (with specific value for $\bar{w}_0$), we record the final learned policy (recommendation probabilities) over the picked items and present the heat map in Figure 4 where darker color represents larger recommendation probability. As shown in Figure 4, with the increase of the value of $\bar{w}_0$, the learned recommendation probabilities of items with large value of dimension 0 (e.g., $i_{15}$ and $i_{16}$) increase while those with small value decrease. Thus, to maximize the platform's commercial benefits, the platform would tend to recommend items that match user's preference over item dimension with high user preference. Similar results can be derived when we analyse how the learned policy adjust to different levels of user uncertainty over item dimensions. Thus, the proposed framework reveals that the policy would adjust more for item dimensions with higher preference or uncertainty so as to maximize the commercial benefits of the platform.

