# OpenReview forum: "PURE: An Uncertainty-aware Recommendation Framework for Maximizing Expected Posterior Utility of Platform"
_ICLR.cc/2021/Conference — Reject_

### Official Review · AnonReviewer3 · 2020-10-27
**The paper aims to model the posterior utility of showing an item given a static display policy, where the utility function captures both: (a) uncertainty over item dimensions from the user perspective, and (b) influence of the policy on the user.**

**Rating:** 5
**Confidence:** 5

**Review:**

The paper aims to model the posterior utility of showing an item given a static display policy, where the utility function captures both: (a) uncertainty over item dimensions from the user perspective, and (b) influence of the policy on the user. It is motivated by the fact that most recommender systems don't take into account that user may be highly uncertain about value/utility in certain dimensions (e.g., color of a product) while more certain about others. The platform can use this information explicitly while optimizing for what to show.

A question that needs empirical evidence is whether uncertainty in certain dimensions have affect on recommendation outcomes. A priori it seems unclear which type of an effect this has. For instance, if a user is more uncertain about some dimensions for item 1 vs item 2, how can the platform exploit it?


 - Isn't the below incorrect as in the very next line you point out works that capture this:

> may affect the users’ behaviors, which is crucial but always ignored.

 - The following seems untrue. There are several papers at RecSys (and workshops that focus on explicit modeling of user behavior and their response to displays, see https://recsys.acm.org/recsys20/workshops/ )

>  However, most existing works in recommendation area do not explicitly model the impact of the display policy
on the user, which might be self-limiting and incomplete.



In eq (1),  the decision variable seems to be a prob. mass function over the items. How is the recommendation performed is unclear at this point.

It is unclear what x represents in Assumption 1/eq (2). The papers say its the representation of the item. So is x a realization of item attributes from the user's perspective? Can they be interpreted as utilities/values per unit?

The normality assumptions in eq (2) and (3): what is the price of reality deviation from these?

It says v_u is the expected item representation given a display policy \pi, but unclear how this can be computed. Assumption 1 does not have any influence of display policy on it.

In eq (3), what is the random variable 'display' indicating? Is it just user imagining that the item i will be displayed? This is very unclear.

The dependence of the utility on display policy is through the vector v_u which is unfortunately not explained fully.

Lemma 1, Corollary 1 and 2 seem to follow in a straightforward manner from standard properties of Gaussians.
The use of importance sampling to deal with a nonlinear f (which is still unclear how we get this in the first place? domain knowledge?) seems straightforward as well.

The use of ADMM procedure, while good, would also be considered standard at this point.

Overall, the paper is well written, although the ideas are relatively less novel.


 - [Minor] Rephrase below:

> .. serving as the role of user aggregation

---

> ### Author Response · Authors · 2020-11-17
> **Response to Reviewer 3**
>
> We sincerely appreciate your detailed comments and address your concerns in the following.
>
> "A question that..., how can the platform exploit it?": Your concern may lie on how the platform could exploit the user uncertainty over different item dimensions to achieve better benefits. For instance, assume that there are two laptops which are similar but with different colors and the target user is uncertain about which color he would prefer. We further assume that the benefits that the platform can gain from recommending these two laptops are different. Then the platform may choose to recommend the laptop with higher benefit by utilizing the user uncertainty over the item color.
>
> "Isn't the below...", "The following seems...": You are absolutely correct that there exist works to explore the impact of recommendation systems. However, we should point out that it is still not well-studied how to explicitly exploit the influence of display policy when developing recommendation solution. One of the purposes of the proposed framework is to provide a novel way to achieve that. We have made this point clearer in the revised paper.
>
> "In eq (1)..., How is the recommendation performed...": Equation 1 describes how to derive the optimal recommendation policy $\pi^u$ for user $u$. The policy $\pi^u$ is a probability distribution over the candidate item set $\mathbb{I}$ which indicates the recommendation probability  for each item. Thus, the platform can perform recommendation by conducting sampling w.r.t. $\pi^u$.
>
> "It is unclear what x represents...": To model the user uncertainty over different item dimensions, we assume that the representation of item $i$ for user $u$ is  sampled from a multivariate Gaussian distribution $p_{u,i}$. Thus, as you said, $x$ can be regarded as a realization of item high-order features from the user's perspective.
>
> "The normality assumptions in eq (2) and (3): what is the price of reality deviation from these?": The proposed framework based on two reasonable assumptions. The first assumption adopts Gaussian distribution to capture the user uncertainty. Such Gaussian-based assumption is widely accepted in the research area. Besides, the second assumption only assumes that higher similarity between item $i'$ and the displayed items would make the user consider that item $i'$ is with higher probability to be displayed, which is also reasonable in real-world scenarios.
>
> "It says v_u..., how this can be computed...": As shown in the paper, $v_u=\sum_{i\in\mathbb{I}}\pi_i\mu_{u,i}$. Please refer to the content below the Equation 3.
>
> "In eq (3), what is the random variable 'display' indicating...": Thanks for the comment. "display" indicates the event of displaying the corresponding item to the target user. We have made that clearer in the revised paper.
>
> "Lemma 1, Corollary 1 and 2 seem to follow in a straightforward manner..., a nonlinear f which is still unclear...": To ease the presentation, we only present the main conclusions in methodology section and place the detailed proofs of Lemma 1, Corollary 1 and 2 in Appendix. We sincerely welcome the readers to read the detailed proofs and provide your suggestions. Besides, as illustrated in the paper, $f$ is the utility function of the platform which can be flexibly defined to fit different requirements of different scenarios. For instance, if we regard the click through rate (CTR) as the platform's utility, $f$ can be an MLP trained to conduct CTR prediction.
>
> We are grateful for your review and have tried our best to address your concerns. It is our duty to make the work clear for the readers and we also hope that the work can be judged with careful consideration. We sincerely look forward to your further comments.

---

### Official Review · AnonReviewer1 · 2020-10-28
**Interesting angle to look at a recommendation problem; but the provided technical details and experiment results are hard to follow**

**Rating:** 4
**Confidence:** 4

**Review:**

This paper studies the problem of system utility optimization by explicitly exploiting user uncertainty regarding a specific item’s properties (e.g., certain dimensions of item representations) and the impact of the display policy. The basic idea is that when a user is uncertain about some properties of a particular item (e.g., quality of the item) or he/she might be more (or less) influenced by the system’s past recommendations, the system would have room to recommend items that maximize system utility. The idea sounds interesting and feasible to me, but the presented technical solution is unfortunately opaque and hard to follow. I do have a few of questions regarding the two assumptions and main results in this work.

First, what is the meaning of $p_{u,i}(x)$? Based on the description, it should be the representation of item $i$ from user $u$’s perspective. However, the way to estimate the parameters of this distribution is based on the system learnt embeddings in user groups and item categories. Why would those learnt embeddings or such an estimation method correspond to a particular user’s assessment on an item? For example, the user group is constructed by age and gender, does it mean all users with the same gender and age would have the same uncertainty on items, disregarding to their specific experiences of those items? I believe this item uncertainty could be better modeled with respect to individual user’s interaction history with the item (e.g., non-durable consumer goods or service might be a better target for this study).

The second assumption is even harder to understand. Should this $p_{u,i}(display|x;\pi)$ be understood as a user’s belief of why item $x$ is recommended to him/her? But how would this estimate on the user side affect his/her behavior? The higher the probability, the more likely the user would click on the result? Based on Eq (5), it seems to be the case; but why would it be in practice? For example, when the user realizes the system is taking advantage of him/her for profit, would the user go against the recommendations? This actually leads to a bigger question regarding the problem setup in this paper: throughout the paper, the tone is the system takes advantage of users to maximize its “commercial benefits”, which is very shortsighted. This is based on a static assumption about users’ behaviors and their trust on the system, as required by Eq (5). However, once the user realizes the system’s strategy and loses his/her trust on the system, the system will lose further opportunities for making recommendations (or gaining its utilities).

The reported experiment results are also not easy to follow. For example, Figure 1 and 2 showed the obtained utility from the learnt policy, which illustrate the learning is converging. But how good is the learnt policy? There should be some baselines for comparison purpose, e.g., at least against a random policy? Figure 3’s result isn’t a surprise: the utility function is a sigmoid function, which is a generalized linear function, over the item representation and user preference. As a result, how the learnt policy reacts to different user preference can be suggested by the linear case’s analysis. Moreover, I do not quite follow the definition of $v_0$: “record the learned expected value (w.r.t. the learned policy) of the elements with dimension 0 of the mean vectors, which is denoted as $v_0$.”

By the way, I could not derive Eq (5) from Eq (4), and this is what I have:
$$
p_{u,i}(x|display,\pi)=\frac{ p_{u,i}(x,display,\pi)}{ p_{u,i}(display,\pi)} =\frac{ p_{u,i}(x) p_{u,i}(display,\pi|x)}{ p_{u,i}(display,\pi)}
$$

**Acknowledgement of author rebuttal** I appreciate the detailed discussions that the authors have provided. However, my concerns still remain, and I am not convinced to improve my overall evaluation.

---

> ### Author Response · Authors · 2020-11-17
> **Response to Reviewer 1**
>
> We sincerely appreciate your detailed comments for making this work better and address your concerns in the following.
>
> "However, the way to estimate the parameters of this distribution...": We appreciate your opinion that there may exist better solution for estimating the parameters of the distribution $p_{u,i}$. However, we should point out that to construct such distribution is not the goal of this paper. We regard $p_{u,i}$ as the input of our framework and provide one practical way to construct it. Any reasonable method to construct $p_{u,i}$ can be adopted within our framework. Besides, we do have considered to construct $p_{u,i}$ by only utilizing  the interaction history of individual user $u$. However, such method faces the challenge of data sparsity as we have to estimate distributions for all items while the interacted items of one user are always limited. That is the reason why we believe that estimating the parameters of the distribution via group-based method is a practical way.
>
> "The second assumption is even harder to understand..., which is very shortsighted...": As presented in Assumption 2, $p_{u,i}(display|x;\pi)$ denotes the display probability of item $i$ from the perspective of user $u$ when the item representation is $x$, taking the impact of display policy $\pi$ over user $u$ into account. Assumption 2 only assumes that higher similarity between the current item representation $x$ and the expected representation of displayed items would lead to higher display likelihood from the perspective of the user. There is no assumption about "users' behaviors and their trust on the system" as you claim. Besides, we might disagree with your opinion that the proposed framework is "shortsighted". Actually, the proposed framework can avoid the shortsighted actions via considering the impact of display policy over user. According to Equation 1, we try to derive a policy to maximize the expected posterior utility while the posterior utility $U_u(i|display;\pi)$ is parameterized by policy $\pi$. In other words, the long-term impact of policy $\pi$ is taken into consideration when we derive the optimal policy, which indeed contributes to avoid shortsighted policy. By the way, such solution is a little bit similar to the reinforcement learning method as both of them consider the future influence when deriving the policy. That is the reason we link our work with reinforcement learning in the related work section.
>
> "Figure 1 and 2..., There should be some baselines...": Thanks for the suggestion. We have addressed this concern according to your suggestion. Please refer to the common response and the Section 4.3 in the revised paper for more details.
>
> "Figure 3's result isn't a surprise..., do not quite follow the definition of $v_0$...": To make it clearer for you, we provide detailed explanation in the following. Assume that the mean representation of items are $\mu_1, \mu_2,...,\mu_m$, then $v_0=\sum_{j=1}^m\pi[j]\mu_j[0]=\mathbb{E}_{j\sim\pi}\mu_j[0]$ where $a[i]$ denote the $i$-th element of vector $a$. Thus, $v_0$ denotes the expected value of item dimension 0 w.r.t. the learned policy and can reflect to what extent the policy has adjusted to make use of item dimension 0. Besides, we should point out that Figure 3 not only present the result of policy adjustment w.r.t the user preference, but also present the result w.r.t. the item variance of the user which is more important as it reveals the influence of user uncertainty.
>
> "I could not derive Eq (5) from Eq (4)...": Note that we take the influence of display policy into account and the event of displaying item $i$ is parameterized by $\pi$. Thus, by adopting Bayes' theorem, we have: $$p_{u,i}(x|display;\pi)=p_{u,i}(x)\frac{p_{u,i}(display|x;\pi)}{p_{u,i}(display;\pi)}$$
>
> Thank you again for your efforts. We have made the work clearer in the revised paper according to all of the four reviewers. Please refer to the revised paper and let us know if you have any further comments.

---

### Official Review · AnonReviewer2 · 2020-10-28
**Interesting approach but lacking sufficient empirical evaluation**

**Rating:** 4
**Confidence:** 1

**Review:**

Summary:

This paper proposes a recommendation approach by optimising expected utility over item display policies. The objective also accounts for the uncertainty of item representations which is particularly interesting. The ADMM method is adopted to search for an approximately optimal item display policy for both linear and nonlinear utility functions. Experiments are conducted and results presented.


Strengths:

The proposed approach incorporates the uncertainty of item representations in recommendation systems which is very interesting in my opinion.

The paper is well organised and it would be even better if lemma 1 and the proof of corollary 2 were moved to Appendix.


Concerns:

I was unable to find any empirical evaluation on test data for the proposed approach, in addition, lacking experiments that compare the proposed method with competing methods is another major concern.

How efficient is the proposed method on large item set (e.g., millions of items)? It seems very challenging to optimise the constrained quadratic program (Eq 21-22) in that case.


Minor comments:

Sec 1 first sentence:
TicTok -> TikTok

In Eq (1) and many other equations:
What is the "display" random variable?

Assumption 1:
the probability of being x -> the density of being x

Page 5 first sentence:
"Corollary 2 reveals that the posterior utility can be effectively calculated within constant time …"
I disagree with this.

Sec 4.1 second paragraph:
"we also incorporate an entropy-based regularization term into our model."
Could this be precise?


Updates: I would like to thank the authors for their response and the updated draft. Unfortunately, I believe the above major concerns are still valid and therefore retain my original rating.

---

> ### Author Response · Authors · 2020-11-17
> **Response to Reviewer 2**
>
> Thanks for your comments. We address your concerns in detail as follows.
>
> "I was unable to find..., lacking experiments...": In the paper, we propose a novel objective function, i.e., the expected posterior utility, to take the influence of display policy over user into account. In our experiments, the expected posterior utility is thus adopted as the metric for conducting evaluation. For the concern of lacking experiments for comparison, we have addressed that in the common response. The supplementary experiments in Section 4.3 demonstrates the superior of the proposed framework. Please refer to the common response and the revised paper for more details.
>
> "How efficient is the proposed method on large item set...": In the worst case, the time complexity of solving Equation 21 and 22 is $\mathcal{O}(N^2)$ where $N$ denotes the number of items. However, note that large-scale recommendation systems in real-world applications such as Taobao always consist of the recall phase and the phase of final ranking. The first phase recalls a small amount of items from millions of items by utilizing efficient approaches.  The second phase always adopts complex but effective methods to conduct final ranking over the filtered items. Thus, the proposed framework can be applied in the second phase and enhances the recommendation system by taking into consideration the user uncertainty over item dimensions and the influence of display policy over user.
>
> "What is the 'display' random variable?": Thanks for your comment. "display" indicates the event of displaying the corresponding item to the target user. We have made that clearer in the revised paper.
>
> Response to other minor comments: We believe all other comments are also addressed in the revised paper. Please refer to the revised paper and let us know if you have any further comments.
>
> Thanks again for your efforts to make this work better. We believe that your major concerns are addressed in the revised paper and look forward to your further comments.

---

### Official Review · AnonReviewer4 · 2020-11-03
**Interesting topic but there remains some concerns on experiments**

**Rating:** 6
**Confidence:** 3

**Review:**

Summary:
This paper studies an interesting problem and proposes a novel recommendation framework to utilize user uncertainty over different item dimensions and the impact of display policy over users. To achieve the maximized expected posterior utility, it also provides a technically sound solution to derive the approximately optimal policy based on ADMM, and achieve provides insights for the commercial recommendation based on the experiments with real-world data.

Reasons for score:
I like the idea of taking into consideration user uncertainty and display policy. My major concern is about experiments (see cons below). Hope the authors can address my concern in the rebuttal period.

Pros:
1. The paper studies the most important problem for recommendation platforms: commercial benefits. The problem itself will have great impacts on real-world scenarios.
2. The proposed recommendation framework is novel for taking into consideration user uncertainty and display policy for maximizing commercial benefits. Deriving the problem into maximizing the posterior utility, this paper also proposes the ADMM-based solution to derive the approximately optimal policy.
3. Experiments on the real-world dataset provide some practical insights about how to achieve commercial benefits for the recommendation platform.

Cons:
1. Although several insightful experiments are provided, I still have several suggestions on the experiments to enhance the quality of the paper:
(1) Although the data is from a real-world scenario, there are only results on expected utility. It might be valuable to investigate the real-world beneficial increases (if it is sensitive for companies to share the exact value, maybe it can provide the value of the increasing percentage).
(2) There is no baseline. It might because this paper utilizes expected posterior utility to evaluate the effectiveness,  this metric may not be suitable for other recommendation baselines. Thus, is it possible to select another metric as a supplement and have several comparison experiments?
2. In the introduction, it would be better to provide more details about “item dimensions” and “user uncertainty”, which seems not very clear to me.

Questions during the rebuttal period:
Please address and clarify the cons above

---

> ### Author Response · Authors · 2020-11-17
> **Response to Reviewer 4**
>
> Thank you for the suggestions to refine the work. We hope that the following reply would address your concerns.
>
> One of your major concern is that there is no baseline in the experiments. We have addressed this concern in the common response. To the best of our knowledge, the proposed framework is the first work that proposes to maximize the expected posterior utility taking the user uncertainty and the influence of display policy into consideration. As a result, there is no suitable framework for conducting straightforward and informative comparison in term of expected posterior utility, as you mentioned in the comments. We also sincerely agree with your suggestion to conduct non-straightforward comparisons in view of the mentioned reason and provide supplementary experiments in the revised paper. Please refer to the common response and Section 4.3 in the revised paper for more details.
>
> Besides, thank you for the suggestion of providing more details about item dimensions and user uncertainty. As typical solutions for recommendation systems always encode each item as an embedding, the item dimensions refer to different dimensions of the item embedding, which can be explained as different high-order features. The user uncertainty measures to what extent the user realizes the item over corresponding dimension such as quality. We have made these clearer in the revised paper.
>
> Thank you again for your comments.

---

### Review · Ethics_Committee · 2021-01-06

**Decision:**

Concerns raised (can publish with adjustment)

**Ethics Review:**

Decision: Concerns raised (can publish with adjustment)

Summary: Reviewers and the Area Chair note the paper's emphasis on exploiting the "information asymmetry" between a recommender system and its users is cause for an ethical review.  The idea in the paper is that a recommender system can utilize the uncertainty of its users in order to maximize the "commercial benefit" of the system.

Factors to consider include:

1. ICLR's Ethical Principles on contributing to society and to human well-being (and in particular “When the interests of multiple groups conflict, the needs of those less advantaged should be given increased attention and priority”).

2. ICLR's Ethical Principles on avoiding harm and taking action to avoid discrimination.

3. Exploiting and manipulating individuals without their knowledge or consent, particularly in ways that they could foreseeably find problematic (e.g., if they knew the full space of options).
---

Optimizing for commercial benefit is not a priori ethically problematic, so the question is whether there are ethical concerns within the paper's focus. As-is, the paper does not put forward a proposal that detracts from human well-being or adds to discrimination. On the other hand, it does not address the potential for exploitation or other foreseeable harms and risks aligned with a commercial focus. Further experiments could be designed and included in the paper, so the interest of the users is also weighted, not just commercial benefits of the platform. The problem could be, for instance, framed both as a maximization of commercial interests and a metric around user interests. We understand however that such work could be outside the scope of this specific work.

In any case, authors would be well advised to note tradeoffs -- including the harms and risks to individuals -- when commercial goals are prioritized. This can include, e.g., loss of wealth, addiction, unequal access to opportunity, polarization. Such documentation would help ground this paper within the landscape of ethical considerations for recommender systems, contributing to the broader goal of understanding harms and risks of commerce-focused systems.

---

### Author Response · Authors · 2020-11-17
**Common Response**

We are sincerely grateful to your comments for making this work better. We have noticed that there is a shared concern that the work may lack experiments for comparison. We would illustrate the reason for that and further present the efforts we have made to address this concern in the following.

We should first point out that our target is to propose a solid framework, which mainly consists of an objective function (i.e., expected posterior utility) and an optimization solution (i.e., the proposed ADMM-based method), to enhance existing recommendation solutions (via taking the user uncertainty over different item dimensions and the impact of display policy over user into consideration). To the best of our knowledge, it is the first complete framework to achieve this target and there is no suitable framework for conducting straightforward and informative comparison in term of expected posterior utility as Reviewer 4 mentioned in the comments. Thus, in the original paper, we focus on verifying the technical effectiveness of the framework in the experiment section.

To further address this concern, we sincerely accept your suggestion to conduct comparison between the learned policy and other heuristic policies such as random policy and prior-based policy. We have conducted supplementary experiments and provide the detailed results in the revised paper which demonstrates that the learned policy achieves great superior over competed policies. Please refer to Section 4.3 in the revised paper for more details.

Thanks again for your comments. We sincerely hope this response would address your concern and look forward to your reply.

---

### Decision · Program_Chairs · 2021-01-07
**Final Decision**

**Decision:**

Reject

**Comment:**

This paper received borderline negative scores. The reviewers all agree that the proposed approach is interesting. However, there are also common concerns around the clarity of the paper, as well as lacking sufficient empirical evaluation. One reviewer also argues that technical contribution is relatively limited. The author responses were taken into account but it didn't manage to swing the reviews. Therefore, I recommend reject and wish the authors can incorporate the feedback in the revision.